# Effects of head alignment devices on working memory and postural support during computer work

**Ju-Yeon Jung[1], Chang-Ki Kang**[1,2,3] *

1 Institute for Human Health and Science Convergence, Gachon University, Incheon, Republic of Korea, 2 Neuroscience Research Institute, Gachon University, Incheon, Republic of Korea, 3 Department of Radiological Science, College of Health Science, Gachon University, Incheon, Republic of Korea

* ckkang@gachon.ac.kr

**Data Availability Statement:** We uploaded the data described in the manuscript to a public repository at the following link: https://www.kaggle.com/datasets/jyjung95/physiology-and-cognition

## Abstract

The most common risk factor of computer workers is poor head and neck posture. Therefore, upright seated posture has been recommended repeatedly. However, maintaining an upright seated posture is challenging during computer work and induces various complaints, such as fatigue and discomfort, which can interfere working performance. Therefore, it is necessary to maintain an upright posture without complaints or intentional efforts during long-term computer work. Alignment devices are an appropriate maneuver to support postural control for maintaining head-neck orientation and reduce head weight. This study aimed to demonstrate the effects of workstations combined with alignment device on head-neck alignment, muscle properties, comfort and working memory ability in computer workers. Computer workers (n = 37) participated in a total of three sessions (upright computer (CPT_U), upright support computer (CPT_US), traction computer (CPT_T) workstations). The craniovertebral angle, muscles tone and stiffness, visual analog discomfort scale score, 2-back working memory performance, and electroencephalogram signals were measured. All three workstations had a substantial effect on maintaining head-neck alignment ($p <$ 0.001), but only CPT_US showed significant improvement on psychological comfort ($p =$ 0.04) and working memory performance ($p =$ 0.024), which is consistent with an increase in delta power. CPT_U showed the increased beta 2 activity, discomfort, and false rates compared to CPT_US. CPT_T showed increased alpha and beta 2 activity and decreased delta activity, which are not conductive to working memory performance. In conclusion, CPT_US can effectively induce efficient neural oscillations without causing any discomfort by increasing delta and decreasing beta 2 activity for working memory tasks.

## Introduction

Computer-based online work environment is getting a lot of attention worldwide during the COVID-19 and as a result, computer related injuries increase dramatically among computer workers [1, 2]. The prevalence of computer related musculoskeletal symptoms is over 53.4% of

**Funding:** This research was supported by a grant from the National Research Foundation of Korea (NRF) grant funded by the Korean government (MSIT) (No. 2020R1A2C1004355; No. 2022R1F1A1062766).

**Competing interests:** The authors declare that they have no known competing financial interests or personal relationships that could have appeared to influence the work reported in this paper.

computer users [3], and 82% spend up to 6 hours a day on a computer [4]. Long lasting computer use causes occupational and/or chronic diseases of the musculoskeletal by inactivity and abnormal postural alignment [5, 6].

Forward head posture (FHP), a representative cause of musculoskeletal disease and pain, is used by most computer workers owing to its high concentration and productivity during work. Therefore, studies have been conducted to prevent the head from moving forward in various ways [7, 8]. By studying the sitting posture of computer workers, reclining and ball-backrest chairs were found to be effective in reducing fatigue and correcting posture, respectively, and improved trunk stability [9, 10]. However, studies have focused only on musculoskeletal function and impairment during ongoing computer work and not on cognitive function and work performance. Previous dynamic workstations for computer users had some limitations that hindered task performance, only helping to improve physical activity but not helping to perform cognitive tasks. Therefore, it is necessary to develop an optimal workstation that considers both musculoskeletal health and cognitive performance [11].

Embodiment cognition studies have reported that cognitive and brain activity changes depend on various postures. In particular, the upright posture is commonly recommended rather than the slouched posture because of its effectiveness in improving self-confidence, arousal, and reducing stress [12, 13]. According to the recent research, upright posture is associated with high-frequency brain activity compared to the slouched posture, which is linked to increased alertness. It has also been reported to be effective in improving word memorization and working memory skills, which require information processing and short-term concentration [14, 15].

However, methods for effectively sustaining an upright posture during long lasting computer work remain insufficient. A tactile biofeedback system may be one of the methods for preventing FHP and maintaining an upright posture during computer work [15, 16], but the user's voluntary participation and conscious efforts are required to correct the upright posture, which causes significant fatigue during work and distracts the user's attention.

Furthermore, a recent study suggested that helping individuals adopt a more relaxed and comfortable posture may be more beneficial for symptom relief than constantly striving for upright sitting. They argued that excessive focus on upright sitting may lead to anxiety and discomfort from an unaccustomed posture [17]. Individuals who were asked to work hard to achieve the correct posture (sitting up straight) to prevent pain experienced more anxiety from avoiding incorrect posture (such as swayback). In addition, sit up straight requires continuous neuromuscular recruitment to control the posture [18, 19]. Thus, constantly maintaining ideal upright posture is difficult when simultaneously performing cognitive task and requires various efforts. Therefore, it is necessary to develop a computer workstation that can contribute to comfort and facilitate maintaining an upright posture without constant effort and discomfort.

In this study, we focused on differences in the effort required for upright posture control, and cervical traction (external assisting alignment device) was employed to increase postural stability with lower motor recruitment. The cervical traction was utilized to increase the base of support and the traction force was changed to apply different support strength by de-weighting a head. The support strength can contribute to ease neuromuscular recruitment by de-weighting a head and decreasing motor variability of head and trunk. Therefore, the application of a traction force that works in the anti-gravity direction can prevent a postural change of tilting downward and can support the maintenance of upright sitting. Furthermore, it also influences to alleviate the conscious burden to maintain an upright posture, which may be induced to concentrate more on cognitive tasks. Furthermore, the cervical traction affects brain activity by modulating the autonomic nervous system. It has been reported that a decreased heart rate occurs due to the activation of the vagal nerve and inhibition of sympathetic nerves during cervical traction (which may be related with delta power increase and

alpha power decrease) [20]. These physiological changes, including increased delta waves and decreased alpha waves, can affect perception, attention, and brain activity [21–24].

A previous study reported that cervical traction had significant effects on preventing FHP during computer work, as well as improving working memory performance [15]. However, they used a cervical traction device to provide tactile feedback and not an upright postural aid. Thus, the effects of mechanical traction on maintaining head-neck alignment and working memory ability remain unclear.

Considering the effect of cervical traction intervention, applying it to an upright sitting posture can affect not only the prevention of FHP, but also the improvement of working condition and cognitive performance. Therefore, the purpose of this study was to demonstrate the effects of upright assistance by various traction strengths on physical (head-neck alignment and muscle properties) and cognitive (comfort and working memory ability) aspects in computer workers, and to determine the most effective cervical traction strength for working memory performance. Therefore, the purpose of this study was to demonstrate the effects of workstations combined with alignment device on head-neck alignment, muscle properties, comfort and working memory ability in computer workers. The hypothesis of this study is as follows: First, for computer workers, workstations equipped with an alignment device are expected to affect head-neck alignment, muscle tone and stiffness, working memory capacity, and comfort. Second, the level of support provided by the alignment device may have varying effects on head-neck alignment, muscle tone and stiffness, working memory ability, and comfort.

## Materials and methods

### Participants

This study used a crossover repeated measures design to evaluate the effects of the workstations. The effect size calculated using G*Power version 3.1.9.4 was 0.283 based on the results of previous studies (Partial η2 = 0.074) on changes in working memory capacity according to posture changes and the sample size was 34 for α error probability of 0.05 and statistical power of 0.95 [25]. Considering a potential dropout rate of 10%, 3 more participants were recruited. Therefore, thirty-seven right-handed healthy computer users (20–30 years old) who used computer for more than 4 hours per day participated in this study after obtaining written informed consent. Participants with a CVA > 50˚ (normal CVA range) were included to avoid the influence of forward head posture. Furthermore, none of the participants had a history of musculoskeletal, neurological, or psychiatric disorders, and participants who had problems affecting computer use, such as headaches and cervical or lumbar pain, were excluded. Therefore, 19 males and 18 females were recruited for this study. All participants were recruited by direct recruitment method using flyers posted in public setting in Incheon, Republic of Korea from 17 February 2021 to 30 December 2021 (Fig 1). Their general characteristics are presented in Table 1. This study obtained prior approval for human subjects research by Gachon University Bioethics Committee institutional review board (IRB No. 1044396-202101-HR-015-01), and word health organization international clinical trials registry platform (WHOICTRP registration number: KCT0007814). All data were collected in Gachon University.

### Experimental protocol and intervention

The environment setting of desk and chair followed a previous study [26]. All experiments were performed in a soundproof room equipped with electroencephalogram (EEG). Before the workstation intervention, the postural angle and muscle properties were measured. Participants performed the 2-back working memory task for 5 min after an adaptation period (5 min) at the workstation for familiarizing themselves with the task. They were instructed to

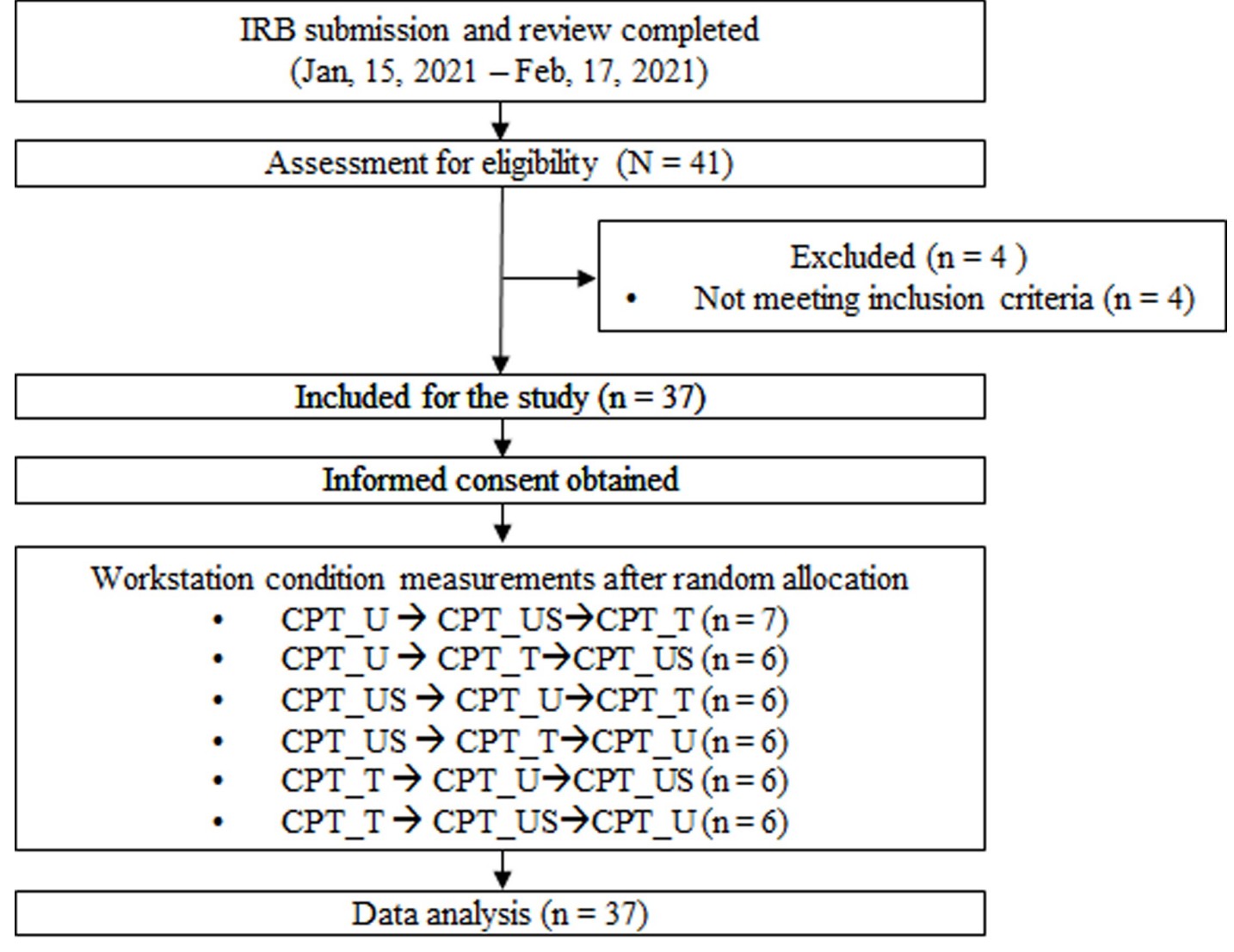

**Fig 1. Flow chart of experimental protocol.**

maintain an upright posture and gaze at the center of the monitor without tilting their heads or keeping their backs on the backrest. EEG signals were measured simultaneously with the 2-back task. During the 2-back task, participants were asked to respond as quickly and

**Table 1. General characteristics of participants.**

|  | Total (n = 37) | Men (n = 19) | Women (n = 18) |
|---|---|---|---|
| Age (years) | 22.3 ± 2.01 | 22.84 ± 2.23 | 21.72 ± 1.56 |
| Height (cm) | 169.79 ± 7.75 | 175.92 ± 5.24 | 162.94 ± 2.78 |
| Weight (kg) | 66.55 ± 11.86 | 73.66 ± 10.06 | 59.04 ± 8.53 |
| CVA (°) | 51.98 ± 7.02 | 52.4 ± 5.93 | 51.53 ± 8.00 |
| Average computer usage period (years) | 11.83 ± 4.29 | 12.35 ± 4.14 | 11.33 ± 4.37 |
| Average computer usage time per day (hours) | 8.41 ± 3.39 | 7.95 ± 3.61 | 8.89 ± 3.07 |

Abbreviations: CVA, craniovertebral angle.

accurately as possible, and they were asked not to move during task performance. All movement was monitored in real-time through an external monitor. After completing the task, they were instructed to maintain the same posture even during the post-test assessment. The Visual Analogue Discomfort Scale (VAS-D), CVA, and muscle properties (tone and stiffness) were measured immediately after the completion of the task. After all measurements were completed, a 5 min break was allowed to remove the effects of the previous intervention. This procedure was applied equally to the next workstation (Fig 2).

To assist in the upright posture, CPT_U used a traction belt placed under the chin without contact to keep participants in an upright position. Therefore, when the participants remained upright, the belt was not in contact; however, when they tilted forward to the monitor, the belt was in contact with the chin and received feedback to move and modify the upright posture again.

CPT_US is a middle-level support workstation that removes head weight load using an alignment device that supports an upright posture with upward traction. The traction force was set to a head weight equal to 7.3% of the individual body weight estimated in a previous study [27]. The participants were asked to relax their neck extensors to determine whether the effects of the head weight support were applied correctly.

CPT_T is a high-level support workstation that assists in maintaining an upright posture by applying a stronger traction force than CPT_US. In CPT_T, the traction force was set to 10% of the body weight. This traction force is commonly applied for therapeutic purposes in neck pain, headache, and spinal deformities [28, 29]. At this traction force, the participants felt that the head was pulled upward, and an upright posture without tilting was maintained without the contribution of the spine extensors. These interventions were performed in random order to eliminate order effects. For increasing compliance of participant, transportation expenses had been provided (Fig 3).

In all workstations, participants wore a chin strap to equally control sensory input for skin contact. The traction belt of CPT_US and CPT_T constantly made contact with their chin from the adaptation period to the end of the post-test, while the traction belt of CPT_U was positioned under their chin without direct contact. Therefore, sensory input for skin contact was equally controlled, with only a difference in supporting strength for each condition.

## Task design

The stimulus displayed on the center of the computer screen. The distance between the participant and the screen was set at 60 cm horizontally from the eyes [30]. One of the nine different

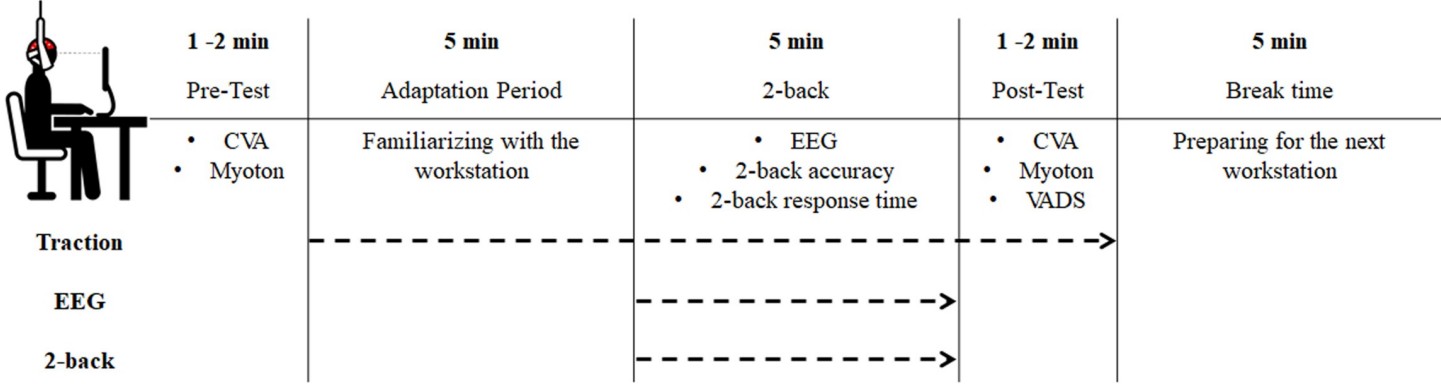

**Fig 2. Diagram of the experimental procedure.**

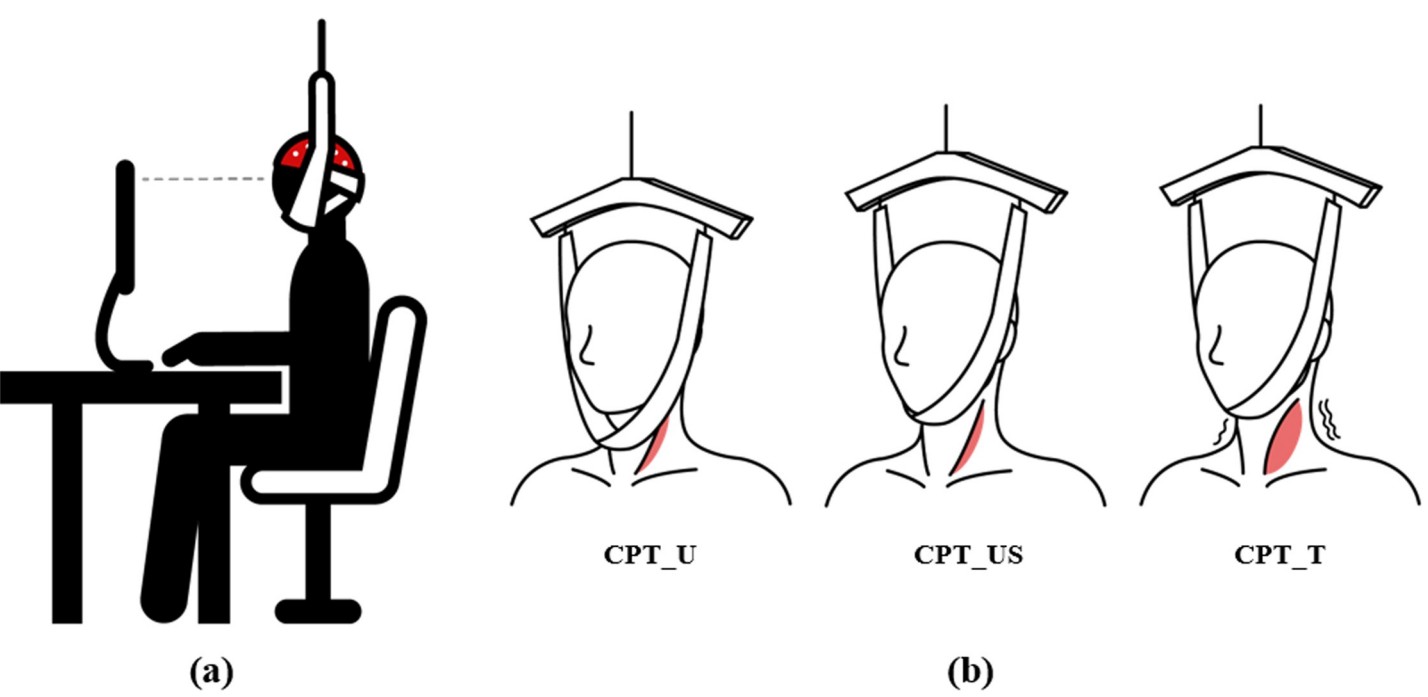

**Fig 3. Task posture and traction computer workstations.** (a) A task posture. (b) Intervention methods of each workstation.

single digits (1–9) in white was used as a stimulus and was randomly displayed individually, and the digit size was set as 30 points font. After the participants were ready for the 2-back working memory task, a countdown of 3 s was given as a starting signal. The stimulus was presented for 500 ms and a cross mark (+) followed by 1500 ms. Thus, each stimulus had a 2000 ms interval and a total of 150 stimuli were given for 5 min. The 2-back working memory task was performed using the DMDX software 2017 (University of Arizona, Tucson, AZ, USA) and the setup followed the previous study [15, 25, 31]. To minimize motion artifacts, participants were instructed not to move except for finger tapping during the task. Consequently, no significant postural instability that could affect cognition was observed.

To evaluate the working memory performance, response time and accuracy (false rate) were collected. The false rate was calculated as the proportion of incorrect responses out of 150 stimulus responses (number of false responses/150 x 100). Response time refers to the average time that the participant responded to stimuli.

### EEG data acquisition and analysis

EEG was performed with 32 active electrodes (QEEG-32Fx, LAXTHA Inc., Daejeon, Korea) at locations based on the 10–20 system. Additionally, four electrodes on the ventral up and horizontal left and right sides of both eyes were used for electrooculogram detection, and two electrodes for electrocardiography were used above and below the left subclavian artery. All signals were recorded using TeleScan software (http://laxtha.net/telescan/) for 5 min during the task period. EEG recordings began at the beginning of the 2-back task and continued for the 5-min task period. All impedances of electrodes were kept below 5kΩ. An online bandpass filter (0.5–50 Hz) was applied during the data digitization and amplification. All the electrodes were referenced online to A1 and A2 (A1 + A2). Based on Chai [32], the 32 channels were classified into prefrontal, frontal, temporal, parietal, and occipital cortices. Considering the effect

of the traction belt located under the occipital condyle, three channels (O1, O2, and Oz) were excluded from the analysis. The traction was consistently maintained throughout the EEG recording.

Independent component analysis (ICA) was performed using MATLAB-based EEGLAB to remove the EOG and ECG components. The brain waves used in the EEGLAB analysis were predefined as follows: First, data for the first 30 s were removed to collect stable EEG signals. Second, the data were re-referenced to the average of all channels without EOG and ECG electrodes using the reference electrode standardization technique (REST) [33]. Finally, for the frequency analysis, a fast Fourier transform was used for the relative spectral power density (%) of the delta (0.5–4 Hz), theta (4–8 Hz), alpha (8–13 Hz), beta (13–30 Hz), beta 2 (15–20 Hz), beta 3 (18–40 Hz), and gamma (30–50 Hz) waves. The general characteristics of each band are presented in Table 2.

## CVA, VAS-D and MYOTON

The CVA is the intersection angle between the horizontal line and the line connecting the C7 spinous process and tragus of the ear. The camera was placed 1.5 m away from the level of the acromion, and the markers were attached to the anatomical landmarks of C7 and the tragus. CVA values were calculated using ImageJ analysis software [38].

The participants responded to their discomfort level in the current state using the VAS-D, which was confirmed immediately after the completion of the 2-back working memory task. The VAS-D score ranged from 0 to 10, with a score of 10 indicating "extreme discomfort" and score 0 is "very comfortable" [39]. VAS-D is a modified assessment of discomfort tool derived from the global assessment VAS (Visual Analogue Scale), which comprehensively evaluates discomfort, pain, fatigue, and other factors. The VAS demonstrates very high reliability with an Intra-class Correlation Coefficients (ICC) of 0.97 [95% CI = 0.96 to 0.98] [40]. According to Harland (2019), VAS-D shows a strong correlation with VAS (r = 0.54, $p<0.001$), and simultaneously reporting a 7.8% higher sensitivity in evaluating discomfort [41].

The tone (Hz) and stiffness (N/m) of the superficial skeletal muscle according to the workstation were measured using a handheld myotonometer with excellent intra and inter-tester reliability (ICC = 0.97) (Myoton AS, Tallinn, Estonia) [42]. Muscle stiffness is the resistance of the myofascial tissue to external force. Muscle tone is vibration caused by passive stretch. The tone and stiffness of all muscles were recorded as average values of three repeated measurements. An experienced physiotherapist measured every outcome, and to minimize biased assessment, the order of interventions was blinded.

The measured muscles included both side of the suboccipital muscles (SM), levator scapulae (LS), upper trapezius (UT), and sternocleidomastoid (SCM). The exact measurement locations were determined based on previous studies [15]. The SM were identified between the midpoint

Table 2. General characteristics of each band.

| Frequency band | Brain states |
|---|---|
| Delta | Selectively suppress non-relevant neural activity [34] |
| Theta | Deeply relaxed, inward focused [35] |
| Alpha | Very relaxed, passive attention [35] |
| Beta | Anxiety dominant, active, external attention [35] |
| Beta 2 | Increases in energy, anxiety [36] |
| Beta 3 | Significant stress, high energy, and high arousal [36] |
| Gamma | Concentration, complex information in consciousness [37] |

of the C2 spinous process and the occiput [43, 44]. The LS were marked just above the superior angle of the scapula [42]. The UT measurement point was marked at the midpoint between the acromion and the spinous process of the seventh cervical vertebra [45]. The SCM was assessed at a point located midway between the manubrium sterni and the mastoid process [46].

## Statistical analysis

An experienced statistical analyzer used Jamovi ver.2.2.5 (https://www.jamovi.org/), and one-factorial repeated-measure ANOVA (RMANOVA) was conducted to test for mean difference (MD) in every outcome. To analyze the differences in CVA, tone and stiffness were analyzed using a four-level (pre-test, CPT_U, CPT_US, CPT_T) RMANOVA. VAS-D, 2-back working memory data (response time and accuracy) and EEG signals were analyzed using a three-level (CPT_U, CPT_US, CPT_T) RMANOVA. Tukey's HSD test was performed for post-hoc pair-wise comparisons, and the standard criterion of statistical significance ($p < 0.05$) was applied for all analyses. The statistically significant outcomes of muscle properties, 2-back working memory data and EEG signals are noted in the Results.

# Results

In this study, a total of 44 participants expressed their willingness to participate, and 37 eligible participants met the study criteria. All participants completed the procedures and assessments without any dropouts or incidents related to accidents or side effects during measurements.

## CVA

CVA measurements were performed to examine the effects of workstations on maintaining an upright alignment during working memory task. As a result, the participants were able to maintain an upright posture at all workstations (CPT_U, 55.28˚; CPT_US, 57.07˚; CPT_T, 55.67˚). All CVA values were above 50˚, which was within the normal alignment range of the CVA, and the CVAs of all workstations were significantly improved compared with the pre-test (F = 7.04; $p < 0.001$). CPT_U, CPT_US, and CPT_T were significantly improved by 3.3˚ (T = -3.01; $p = 0.024$), 5.09˚ (T = -3.79; $p = 0.003$), and 3.69˚ (T = -2.97; $p = 0.027$), compared to the pre-test, respectively. However, no significant differences were observed between the workstations (Table 3).

**Table 3. CVA differences by workstation.**

| Repeated Measure ANOVA | | | | | | Post Hoc Comparisons (Tukey) | | | |
|---|---|---|---|---|---|---|---|---|---|
| Dependent Variable | Fixed Factors | Mean ± SD | F | p | $\eta_p^2$ | Variables | | T | p |
| CVA (˚) | pre-test CPT_U CPT_US CPT_T | 51.98 ± 7.02 55.28 ± 5.01 57.07 ± 5.56 55.67 ± 6.59 | 7.04 | <0.001* | 0.172 | pre-test | CPT_U | -3.01 | 0.024* |
| | | | | | | | CPT_US | -3.79 | 0.003* |
| | | | | | | | CPT_T | -2.97 | 0.027* |
| | | | | | | CPT_U | CPT_US | -2.10 | 0.172 |
| | | | | | | | CPT_T | -0.39 | 0.980 |
| | | | | | | CPT_US | CPT_T | 1.19 | 0.636 |

Abbreviations: CVA, craniovertebral angle; CPT, computer; CPT_U, upright CPT workstation; CPT_US, upright support CPT workstation; CPT_T, traction CPT workstation; $\eta_p^2$, partial eta-squared; SD, standard deviation.

* Statistically significant difference: $p < 0.05$

### VAS-D

Discomfort changed significantly according to the workstation (F = 4.00; $p$ = 0.023). The discomfort was the lowest in CPT_US, with a mean VAS-D of 3.03, which was significantly lower by 0.84 than CPT_U (T = 2.55; $p$ = 0.04) (Table 4).

### Muscle properties

Traction intervention with three different workstations for assisting the upright posture had a significant effect on the mechanical properties of the neck muscles. In CPT_T, left suboccipital muscle tone (Tone_L_SM) was increased significantly by 1.02 Hz compared to the pre-test (CPT_T vs pre-test: $p$ = 0.004), and by 0.78 Hz compared to CPT_U (CPT_U vs CPT_T: $p$ = 0.045). However, there were no differences among the pre-test, CPT_U, and CPT_US in Tone_L_SM (Table 5).

The right sternocleidomastoid tone (Tone_R_SCM) significantly increased at all three workstations compared with the pre-test (pre-test vs CPT_U: MD = 0.5 Hz; $p$ = 0.009) (pre-test vs CPT_US: MD = 0.53 Hz; $p$ = 0.005) (pre-test vs CPT_T: MD = 0.82 Hz; $p<0.001$). In addition, Tone_R_SCM of CPT_T was also significantly increased by 0.28 Hz than CPT_US (CPT_T vs CPT_US: $p$ = 0.006). However, no significant differences were observed between the CPT_U and CPT_US groups (Table 5).

Right sternocleidomastoid stiffness (Stiffness_R_SCM) showed a significant increase in CPT_T (CPT_T vs pre-test: MD = 15.62 N/m; $p<0.001$) (CPT_T vs CPT_U: MD = 10.76 N/m; $p<0.001$). However, there was no significant difference between the pre-test, CPT_U, and CPT_US (Table 5).

### 2-back working memory performance

The performance of the 2-back working memory task was significantly affected by the workstation. First, the mean false rate was the lowest for CPT_US (13.32%) ($p$ = 0.034) and it was significantly lower by 2.05% than CPT_U ($p$ = 0.024). However, the response time was not significantly different between the workstations ($p$ = 0.364) (Table 6).

### Relative spectral power

In the delta wave, a total of twelve channels including whole brain regions changed significantly, nine of which appeared in the prefrontal and frontal lobes (Fig 4). In the prefrontal lobe, seven channels (Fp1, AF3, AF4, AFz, F3, F4, and F8) changed depending on workstation. The largest increase between CPT_US and CPT_T was appeared in AF3 (MD = 9.14; $p$ = 0.005).

**Table 4. VAS-D differences by workstation.**

| Repeated Measure ANOVA | | | | | | Post Hoc Comparisons (Tukey) | | | |
|---|---|---|---|---|---|---|---|---|---|
| Dependent Variable | Fixed Factors | Mean ± SD | F values | p | $\eta_p^2$ | Variables | | T | p |
| VAS-D | CPT_U | 3.87 ± 2.03 | 4.00 | 0.023* | 0.103 | CPT_U | CPT_US | 2.55 | 0.040* |
| | CPT_US | 3.03 ± 1.86 | | | | | CPT_T | 2.07 | 0.111 |
| | CPT_T | 3.11 ± 1.70 | | | | CPT_US | CPT_T | -0.10 | 0.994 |

Abbreviations: CPT, computer; VAS-D, visual analog discomfort scale; CPT_U, upright CPT workstation; CPT_US, upright support CPT workstation; CPT_T, traction CPT workstation; $\eta_p^2$, partial eta-squared; SD, standard deviation.

* Statistically significant difference: $p<0.05$

**Table 5. Muscle properties differences by workstation.**

| Repeated Measure ANOVA | | | | | | Post Hoc Comparisons (Tukey) | | | |
|---|---|---|---|---|---|---|---|---|---|
| Dependent Variable | Fixed Factors | Mean ± SD | F | p | $\eta_p^2$ | Variables | | T | P |
| Tone_L_SM (Hz) | | | 6.65 | 0.001* | 0.160 | pre-test | CPT_U | -1.66 | 0.359 |
| | pre-test | 16.73 ± 1.98 | | | | | CPT_US | -2.08 | 0.181 |
| | CPT_U | 16.97 ± 1.60 | | | | | CPT_T | -3.68 | 0.004* |
| | CPT_US | 17.30 ± 1.72 | | | | CPT_U | CPT_US | -1.12 | 0.679 |
| | CPT_T | 17.75 ± 2.05 | | | | | CPT_T | -2.75 | 0.045* |
| | | | | | | CPT_US | CPT_T | -2.3 | 0.118 |
| Tone_R_SCM (Hz) | | | 13.58 | <0.001* | 0.274 | pre-test | CPT_U | -3.37 | 0.009* |
| | pre-test | 12.49 ± 0.89 | | | | | CPT_US | -3.62 | 0.005* |
| | CPT_U | 12.99 ± 0.75 | | | | | CPT_T | -5.77 | <0.001* |
| | CPT_US | 13.02 ± 0.79 | | | | CPT_U | CPT_US | -0.28 | 0.992 |
| | CPT_T | 13.31 ± 0.73 | | | | | CPT_T | -2.52 | 0.074 |
| | | | | | | CPT_US | CPT_T | -3.54 | 0.006* |
| Stiffness_R_SCM (N/m) | | | 8.21 | <0.001* | 0.186 | pre-test | CPT_U | -1.22 | 0.620 |
| | pre-test | 194.16 ± 26.36 | | | | | CPT_US | -2.43 | 0.090 |
| | CPT_U | 199.03 ± 19.29 | | | | | CPT_T | -4.18 | <0.001* |
| | CPT_US | 203.57 ± 20.46 | | | | CPT_U | CPT_US | -1.63 | 0.373 |
| | CPT_T | 209.78 ± 20.79 | | | | | CPT_T | -4.19 | <0.001* |
| | | | | | | CPT_US | CPT_T | -2.67 | 0.052 |

Abbreviations: CPT, computer; CPT_T, traction CPT workstation; CPT_U, upright CPT workstation; CPT_US, upright support CPT workstation; $\eta_p^2$, partial eta-squared; SD, standard deviation; Stiffness_R_SCM, right sternocleidomastoid stiffness; Tone_L_SM, left suboccipital muscle tone; Tone_R_SCM, right sternocleidomastoid tone.

* Statistically significant difference: $p < 0.05$

In the dorsolateral prefrontal region, the F3 and F4 delta waves were significantly higher in CPT_US than in CPT_T (MD in F3 = 5.05; $p = 0.023$) (MD in F4 = 6.2; $p = 0.01$). In addition, the F8 of ventrolateral prefrontal regions and the FC5 and Cz of frontal lobe were also significantly increased on the equal contrast (Fig 4 and S1 Table).

The alpha waves changed significantly in the prefrontal and temporal lobes (Fig 4). In the prefrontal region, the alpha waves of the two channels, Fp1 and AF3, increased significantly in the CPT_T compared to CPT_U (MD of Fp1 = 2.87; $p = 0.004$ and MD of AF3 = 3.06;

**Table 6. False rate differences by workstation.**

| Repeated Measure ANOVA | | | | | | Post Hoc Comparisons (Tukey) | | | |
|---|---|---|---|---|---|---|---|---|---|
| Dependent Variable | Fixed Factors | Mean ± SD | F | p | $\eta_p^2$ | Variables | | T | P |
| False rate (%) | CPT_U | 15.38 ± 6.91 | 3.54 | 0.034* | | CPT_U | CPT_US | 2.76 | 0.024* |
| | CPT_US | 13.32 ± 7.85 | | | 0.09 | | CPT_T | -0.42 | 0.907 |
| | CPT_T | 15.87 ± 7.61 | | | | CPT_US | CPT_T | -2.32 | 0.066 |
| Response time (ms) | CPT_U | 653.70 ± 189.53 | 0.97 | 0.364 | 0.026 | - | - | - | - |
| | CPT_US | 677.81 ± 197.39 | | | | - | - | - | - |
| | CPT_T | 663.60 ± 221.12 | | | | - | - | - | - |

Abbreviations: CPT, computer; CPT_U, upright CPT workstation; CPT_US, upright support CPT workstation; CPT_T, traction CPT workstation; $\eta_p^2$, partial eta-squared; SD, standard deviation.

* Statistically significant difference: $p < 0.05$

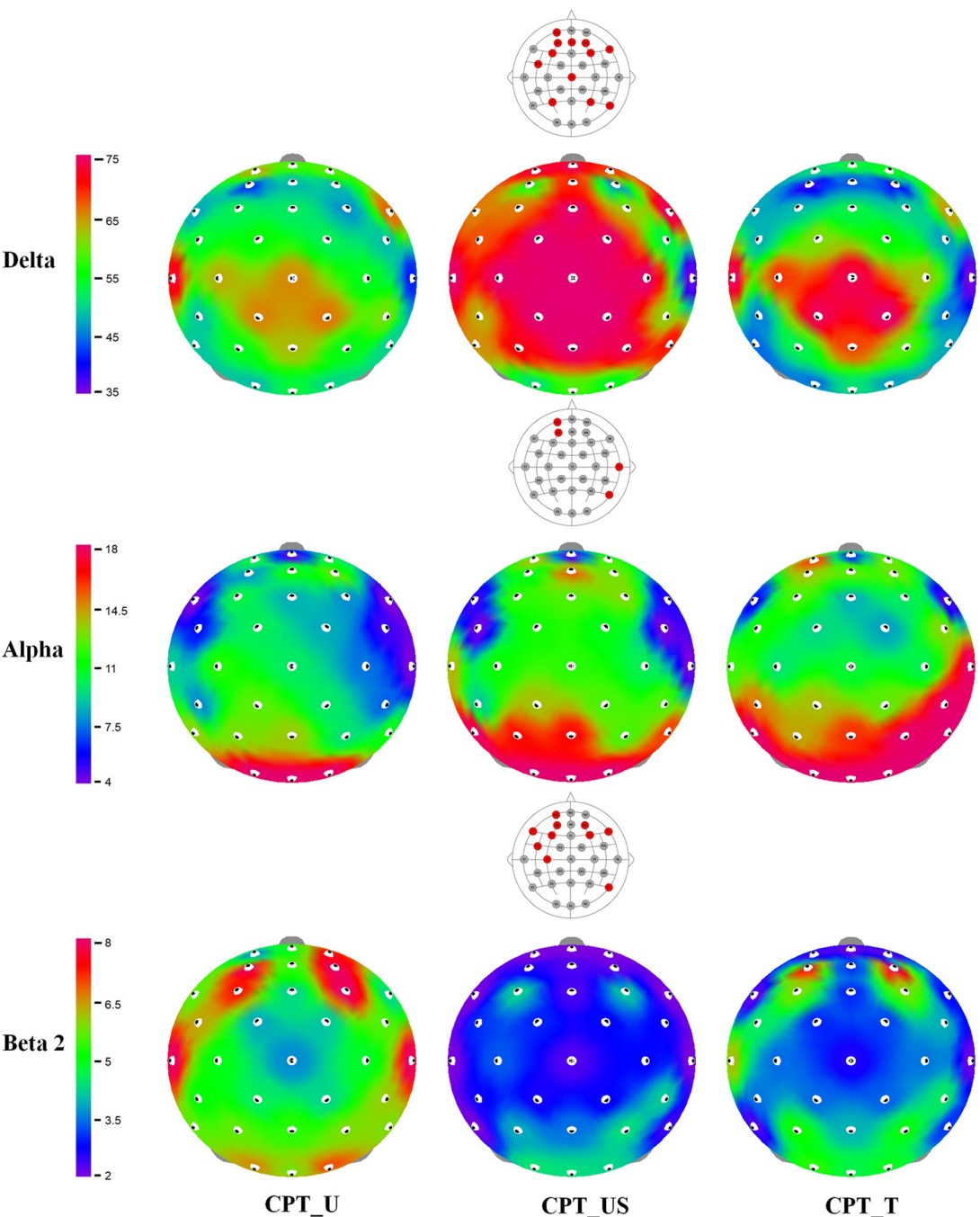

**Fig 4. The topographies for the differences between intervention methods in the delta, alpha and beta 2 spectral power.** The EEG channels with significantly differences represents with red dots on upper panel respectively. The color bar indicates the power spectrum value of each channel, where a higher value is represented by the red color.

$p = 0.016$) and CPT_US (MD of Fp1 = 1.9; $p = 0.049$ and MD of AF3 = 2.07; $p = 0.046$). In the temporal region, the alpha waves of T8 increased significantly in CPT_T compared to CPT_U (MD = 2.49; $p = 0.018$) and CPT_US (MD = 1.97; $p = 0.034$), whereas in P8, it increased by 2.76, compared to CPT_US ($p = 0.014$). However, there was no significant difference between CPT_U and CPT_US in the alpha waves (Fig 4 and S2 Table).

Beta 2 showed a significant decrease in the CPT_US in the whole brain regions (Fig 4). In the prefrontal region, the beta 2 waves of seven channels (Fp1, AF3, AF4, F3, F4, F7, and F8) showed significant differences. The largest decrease between CPT_US and the CPT_U was appeared in AF3 (MD = 1.22; $p$ = 0.036). The channels with the largest differences between the CPT_US and the CPT_T was AF4 (MD = 1.36; $p$ = 0.004). In F4, it was significantly decreased by 1.16, compared to CPT_U ($p$ = 0.031) and by 0.71, compared to CPT_T ($p$ = 0.032).

The significant decreases between CPT_US and the CPT_U were appeared in FC5 of frontal lobe (MD = 0.87; $p$ = 0.004), P8 of temporal lobe (MD = 0.78; $p$ = 0.049), and C3 of the parietal lobe (MD = 0.57; $p$ = 0.042). There were no significant differences between CPT_U and CPT_T in the beta 2 waves (Fig 4 and S3 Table).

## Discussion

### Maintenance of an upright posture by workstations

To exclude the influence of postural alignment, alignment devices were applied to maintain an upright posture. CPT_U, CPT_US, and CPT_T are designed to assist in maintaining upright posture during computer work. To confirm the maintenance of an upright posture, CVA assessment was performed. The higher CVA values (closer to 90°) indicated better upright alignment. As a result, CVA improvement was appeared for all CPT_U, CPT_US, and CPT_T compared to the pre-test (51.98°) after performing 5 minutes of the working memory task, and there were no significant differences in CVA among CPT_U, CPT_US, and CPT_T. Considering that CVA typically decreases by forward lean during computer work, traction workstations showed significant effects in preventing forward lean and maintaining an upright posture. Therefore, this study found that all three workstations for upright assistance had significant effects on maintaining an upright alignment during working memory task.

### Difference between VAS-D by workstations

The workstations showed significant influence on participants' psychological mood (comfort). In this study, only CPT_US showed significantly improved comfort compared to CPT_U. This suggests that increased support strength in CPT_US compared to CPT_U may influence to comfort. The highest discomfort was observed in CPT_U, indicating that maintaining an upright posture during task is difficult and can induce considerable discomfort. In addition, traction intervention in CPT_T seems to be unfamiliar to subjects who have not experienced the alignment device during task, so this novel workstation can cause discomfort. However, CPT_US had a significant effect on improving comfort during working memory task because it does not provide overload by the traction force, but also helps with an upright posture.

### Effect of muscle properties by workstations

The tone of the SCM increased with the assistance level. This change appears to occur because the tone of the SCM are affected by CVA improvement [47]. We found that CVA was also increased in all three workstations compared to the pre-test, as was the tone in the SCM. It is known that exercises for cervical alignment activate the cervical flexors (e.g., SCM) [48].

Furthermore, the SCM originates in the manubrium and is inserted into the mastoid process, it can be affected by an upward traction force on the head. The CPT_T traction force also affected the Stiffness_R_SCM. In CPT_T, it was significantly increased compared to the pretest but did not increase in CPT_U and CPT_US. That is, SCM stiffness was not affected by CVA (unlike tone), but passive stretching due to strong traction could increase SCM stiffness.

The tone of the SM was also affected by strong traction. We found a significant increase in Tone_L_SM in CPT_T compared to the pre-test. As the muscle fiber direction of SM can be affected by an upward traction force. This is known to be a contraction activity to prevent over-stretching of the muscle induced by strong traction force [20]. Therefore, we confirmed that strong traction during the CPT_T increased the tone and stiffness of the SCM and SM, but those of the LS and UT muscles were unaffected. This may be because the LS and UT are less affected by the traction force owing to the muscle fiber direction [49].

## Difference 2-back task performance by workstations

Working memory performance improved more in CPT_US than CPT_U. The mean false rate of CPT_US was the lowest among the three workstations and was significantly lower than that of CPT_U. The decrease in the false rate of 2-back task in CPT_US compared to CPT_U indicates the close relation between comfort and working memory performance enhancement. This comfort feeling has lots of influence on cognitive attention. According to discomfort studies, the uncomfortable experience can lead performance deficits due to leading attention directed away from the task. This distraction may lead to slower responses or slower cognitive processing in task-relevant regions (decrease in cognitive performance) [50, 51]. Therefore, comfortable workstation can have a significant impact on working memory performance. However, the response time was not affected by the workstations. Unlike this study, Jung (2022) argued that the poorly posture (forward head) contributes to slower responses in working memory performance [15]. Thus, we excluded the influence of poorly posture alignment by workstations which make participants into an upright alignment and minimize the impact of posture alignment. In this study, there were no significant differences in CVAs, suggesting that the influence of poorly posture was effectively controlled unlike the previous research. Therefore, the false rate appeared to be highly related to postural support, whereas the response time appeared to be related to postural alignment.

## Effect of EEG signals by workstations

The delta activity was significantly higher during the CPT_US than CPT_T. The 75% of significant channels were observed in the prefrontal and frontal regions. This result is in agreement with previous reports of increased delta activity in frontal regions [34]. In addition, the increased delta activity during the "continuous performance task" reflects the association with sustained attention [52]. Likewise, sustained attention during 5 min of working memory task may cause a delta power increase.

The comfort level of CPT_US can also affect the increase in delta power. Delta waves can be promoted in physiologically and psychologically relaxed states [53]. The existing evidence shows that relaxation is characterized by increased low frequency oscillations. The lower-frequency EEG bands (including delta and theta band) of the frontal pole may indicate more relaxing or comfortable environments and may be associated with positive subjective moods [54]. However, in CPT_T, the overload of assistance may have applied as excessive stimulus during the task and reduce delta wave activity. Pan et al. (2012) reported that neck tightness and discomfort increased as traction device strength increased [20]. Therefore, we concluded that CPT_US improved comfort level compared to the CPT_U (Table 4), and delta waves can be promoted in physiologically and psychologically relaxed states.

The alpha wave activity is affected by memory load or neuronal information processing. The increase in alpha activity in frontal areas with increasing memory load in a working memory task was interpreted as the active inhibition of neural networks for preventing disturbing information during working memory tasks [55]. CPT_T showed significantly increased alpha

wave activity in the prefrontal and temporal regions. The prefrontal alpha wave could be interpreted as the highly active inhibition required for CPT_T compared to the other workstations because of the more disturbing information caused by the high-intensity traction force.

The enhancement of temporal (T8, P8) alpha waves originating in the somatomotor areas can be explained by the inhibition of the somatomotor system [55]. The hyper tone of the SCM in CPT_T stimulates somatomotor areas and affects alpha activity. Therefore, the increased alpha activity in the CPT_T indicates that the high traction force may act as an interfering stimulation that should be suppressed in the working memory process.

A significant reduction in beta 2 activity was observed in the CPT_US group compared to the CPT_U or CPT_T. In CPT_U and CPT_T, the increased beta 2 activity occurred in the prefrontal, frontal, and parietal regions which were important to the working memory process, such as forming memory, maintaining the current contents, and preventing interference from distraction [56, 57]. In particular, the activity is greatly increased while maintaining continuous muscle contraction, and it is known that activity increases in postural control and contraction [58, 59]. In CPT_U, because the participant consciously maintained an upright posture, the imposed motor control may influence to the increase of the beta 2 activity in the motor cortex (FC5, C3). In the CPT_T, contraction by passive stretching of the SM and SCM due to strong traction seemed to have affected the increase in beta 2. This assumption was also explained by the decreased beta 2 activity in CPT_US. The CPT_US induced lower motor control activity and relaxed the postural muscles, and the ease of motor control was related to lower beta 2 activity in the motor cortex.

## Conclusion

We found the effort to maintain an upright posture in CPT_U causes discomfort, resulting in increased false rate and beta 2 activity in working memory performance. In CPT_T, the high traction force significantly changed the muscle properties of the SM and SCM. This change could cause increased alpha and beta 2 activity. In CPT_US, the participants were able to maintain an upright posture without any effort, and the traction force did not influence the mechanical properties. This could cause increased comfort and working memory performance, as demonstrated by the EEG signals of increased delta activity and decreased beta 2 activity. Using the CPT_US, this study observed that working memory improved with proper postural posture assistance by alignment device.

The limitations of this study were as follows. First, it is difficult to confirm the long-term effects of the workstations because the present study applied a 5 min single intervention at each workstation. To generalize this study to an actual work environment, further studies on intervention time or frequency are needed. Second, although CPT_T affected EEG activity and muscle properties by high traction force in the healthy group, the positive effect of traction was expected to be pronounced in the patient group, such as those with alignment disorders or work-related musculoskeletal disorders. Thus, the current investigation may not reflect the outcomes that might occur in other populations. As a result, the external validity requires careful consideration when applying them to diverse groups.

## Supporting information

**S1 Table. Repeated measure analysis results of delta wave relative spectral power.**
(DOCX)

**S2 Table. Repeated measure analysis results of the alpha wave relative spectral power.**
(DOCX)

**S3 Table. Repeated measure analysis results of beta 2 waves relative spectral power.** (DOCX)

## Author Contributions

**Conceptualization:** Ju-Yeon Jung, Chang-Ki Kang.

**Data curation:** Ju-Yeon Jung.

**Formal analysis:** Ju-Yeon Jung, Chang-Ki Kang.

**Funding acquisition:** Chang-Ki Kang.

**Investigation:** Ju-Yeon Jung.

**Methodology:** Ju-Yeon Jung, Chang-Ki Kang.

**Project administration:** Chang-Ki Kang.

**Resources:** Ju-Yeon Jung.

**Software:** Ju-Yeon Jung.

**Supervision:** Chang-Ki Kang.

**Validation:** Ju-Yeon Jung.

**Visualization:** Ju-Yeon Jung.

**Writing – original draft:** Ju-Yeon Jung.

**Writing – review & editing:** Chang-Ki Kang.

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
