## [Decision Letter · Decision Letter 0]

26 Dec 2023

PONE-D-23-37254Effect of a postural assistance to improve cognitive productivity and physical health during computer work - EEG studyPLOS ONE

Dear Dr. Kang,

Thank you for submitting your manuscript to PLOS ONE. After careful consideration, we feel that it has merit but does not fully meet PLOS ONE’s publication criteria as it currently stands. Therefore, we invite you to submit a revised version of the manuscript that addresses the points raised during the review process.

We appreciate your submission of the manuscript titled "Effect of a postural assistance to improve cognitive productivity and physical health during computer work - EEG study" to PLOS ONE. After a thorough review by two reviewers, it has been determined that the introduction, methods, and results sections require major revisions to provide additional details and enhance clarity, as per their precise assessment.

We look forward to receiving your revised manuscript.

Kind regards,

Mohammadreza Pourahmadi, PT, Ph.D., Postdoctoral Fellow

Academic Editor

PLOS ONE

 [This research was supported by a grant from the National Research Foundation of

Korea (NRF) grant funded by the Korean government (MSIT) (No. 2020R1A2C1004355; No. 2022R1F1A1062766).].  

3. In the online submission form, you indicated that [Insert text from online submission form here]. 

4. PLOS requires an ORCID iD for the corresponding author in Editorial Manager on papers submitted after December 6th, 2016. Please ensure that you have an ORCID iD and that it is validated in Editorial Manager. To do this, go to ‘Update my Information’ (in the upper left-hand corner of the main menu), and click on the Fetch/Validate link next to the ORCID field. This will take you to the ORCID site and allow you to create a new iD or authenticate a pre-existing iD in Editorial Manager. Please see the following video for instructions on linking an ORCID iD to your Editorial Manager account: https://www.youtube.com/watch?v=_xcclfuvtxQ.

6. We note that Figure 2 and 3 in your submission contain copyrighted images. All PLOS content is published under the Creative Commons Attribution License (CC BY 4.0), which means that the manuscript, images, and Supporting Information files will be freely available online, and any third party is permitted to access, download, copy, distribute, and use these materials in any way, even commercially, with proper attribution. For more information, see our copyright guidelines: http://journals.plos.org/plosone/s/licenses-and-copyright.

a. You may seek permission from the original copyright holder of Figure 2 and 3 to publish the content specifically under the CC BY 4.0 license. 

Reviewers' comments:

Reviewer's Responses to Questions

**Comments to the Author**

1. Is the manuscript technically sound, and do the data support the conclusions?

Reviewer #1: Yes

Reviewer #2: Partly

2. Has the statistical analysis been performed appropriately and rigorously? 

Reviewer #1: Yes

Reviewer #2: Yes

3. Have the authors made all data underlying the findings in their manuscript fully available?

Reviewer #1: Yes

Reviewer #2: Yes

4. Is the manuscript presented in an intelligible fashion and written in standard English?

Reviewer #1: Yes

Reviewer #2: Yes

5. Review Comments to the Author

Reviewer #1: The study is interesting and meaningful. People spend large amount of time in working on computer.

Just a minor comment:

How long is the adaptation period at the workstation?

It is suggested to use a chart or diagram to illustrate the experimental protocol with the timeline.

Reviewer #2: The purpose of the study was to examine the effect of head alignment on the cognitive productivity and physical health of computer workers. The study is interesting but needs clarification. Some comments are below.

Comments:

Title – Please, consider revising the title. It is not linked to the rationale of the manuscript. The term “assistance” seems not to be appropriate to represent the effect of postural orientation on cognitive tasks. It is confusing. The terms “productivity” and “physical healthy” are generic. The title should reflect the main findings/message of the study.

Abstract – this section needs revision. Instead of asking a question, it is necessary to clarify the meaning of postural assistance to the purpose of the study. The statement “all workstations had a substantial effect on postural assistance” needs context to understand. In general, the letter p for statistics is always represented in lowercase. Please, clarify what “physical stability” means. Instead of being generic with “brain dynamics”, please, include at least one explanation about the brain activation results based on frequency analyses. What is the main point and message?

Introduction – this section addresses the usability of workstations for computer workers focusing on musculoskeletal impairments and reduced cognitive performance. Line 65 – Please, clarify what components of physical health and cognitive performance could be impaired by postural orientations (e.g., upright sitting). How postural stabilization can influence cognitive performance? Line 75 – It is necessary to define the concepts of physical health and cognitive productivity, as their meanings are unclear at present. Lines 77-84 – The introduction did not provide sufficient background to understand the relationship between postural stability, alignment devices (cervical traction), and cognitive performance. Line 80 – Please, be more specific in the definition of “physical and cognitive aspects” and “effective strength for productivity and safety at work” (line 81). As said, the title, abstract, and introduction sections seem not to be linked. The purpose of the study presented in the abstract seems different than the introduction section. Please, provide the hypotheses for all variables mentioned in the last paragraph of this section (lines 81-84).

Materials and Methods – Lines 87-88 – the first sentence of this section is odd. Lines 88-90 – Please, provide the reference for sample size calculation. Line 106 – Please, provide the minimum information to understand the experimental setup without being conducted to another study. Please, provide a figure or picture of the experimental procedures. What were the instructions provided to the participants? Line 111 – Please, specify muscle properties. Were the participants in a sitting position during the familiarization period? Line 118-122 – in the CPT_US condition, does the participant was in constant contact with the traction device? Considering that postural control is influenced by sensorial information and moments of postural instability seem to affect the brain activation, please, provide information on how those systems were affected by each condition. What is being manipulated by these conditions? Please, provide information about Figure 1 and Figure 2 e replace when they first appear since they are not related to participants' payment. Task design – Were the numbers displayed on the computer screen? What was the distance between the participants and the screen? "Why was the duration of the stimulus interval so lengthy?" Some studies regarding postural control have shown that stimulus duration and longer intervals may affect postural stability. What instructions were given to the participants? What was the outcome of the cognitive task performance? How the EEG data acquisition was synchronized with the cognitive task and traction devices. Lines 160-162 – Please, provide the rationale to describe brain activation in terms of frequency bands and the meaning of each band for the study. Is VADs a validated test? Lines 173-179 – what were the specific variables for tone and stiffness? Overall, please, provide a complete description of the experimental setup.

Results – Lines 198-200, 207-209, and others, - Please report the statistics result adequately (F values, p values). For example, in general, the letter p for statistics is always represented in lowercase. Include the effect size values for each analysis. N-back results are unclear. Please, describe how the variables (false rate and response time) were computed in the Methods section. Table 5 – Please, provide the time response of the working memory task in each workstation condition. Figure 3 – please, provide the meaning of the colors.

Discussion – A different purpose seems to be stated in the first sentence. Lines 285-287 – It is necessary to further explore the relationship between an upright posture, CVA, and enhancements in working memory task performance based on the results of the study. Line 310-317 – the implications of the posture alignments and working memory outcomes are poorly discussed. Please, discuss how head position and additional somatosensory information (traction) affect the postural stability and how it influences the performance of the working memory task. Does postural alignment affect working memory performance including false rate and response time due to the use of the same CNS resources? Lines 325-326 – Could you please clarify the evidence that supports this statement? There is no mention of physical health and no hypotheses related to it. Please, revise this concept and bring the hypotheses and discussion about this issue.

6. PLOS authors have the option to publish the peer review history of their article (what does this mean?). If published, this will include your full peer review and any attached files.

Reviewer #1: No

Reviewer #2: No

---

## [Author Response · Author response to Decision Letter 0]

4 Mar 2024

The responses to the journal requirements, first reviewer, and second reviewer have been described in the Word file titled "20240222_Response to Reviewers.docx". Please find the attached response file.

---

## [Decision Letter · Decision Letter 1]

26 Jun 2024

Effects of head alignment devices on working memory and postural support during computer work

PONE-D-23-37254R1

Dear Dr. Chang-Ki Kang,

We’re pleased to inform you that your manuscript has been judged scientifically suitable for publication and will be formally accepted for publication once it meets all outstanding technical requirements.

Kind regards,

Ravi Shankar Yerragonda Reddy, Ph.D

Academic Editor

PLOS ONE

Reviewers' comments:

Reviewer's Responses to Questions

**Comments to the Author**

1. If the authors have adequately addressed your comments raised in a previous round of review and you feel that this manuscript is now acceptable for publication, you may indicate that here to bypass the “Comments to the Author” section, enter your conflict of interest statement in the “Confidential to Editor” section, and submit your "Accept" recommendation.

Reviewer #1: All comments have been addressed

Reviewer #2: All comments have been addressed

2. Is the manuscript technically sound, and do the data support the conclusions?

Reviewer #1: Yes

Reviewer #2: Yes

3. Has the statistical analysis been performed appropriately and rigorously? 

Reviewer #1: Yes

Reviewer #2: Yes

4. Have the authors made all data underlying the findings in their manuscript fully available?

Reviewer #1: Yes

Reviewer #2: Yes

5. Is the manuscript presented in an intelligible fashion and written in standard English?

Reviewer #1: Yes

Reviewer #2: Yes

6. Review Comments to the Author

Reviewer #1: (No Response)

Reviewer #2: The authors have made significant improvements to the manuscript, making it suitable for publication.

7. PLOS authors have the option to publish the peer review history of their article (what does this mean?). If published, this will include your full peer review and any attached files.

Reviewer #1: No

Reviewer #2: No

---

## [Editor Report · Acceptance letter]

2 Jul 2024

PONE-D-23-37254R1 

PLOS ONE

Dear Dr. Kang, 

I'm pleased to inform you that your manuscript has been deemed suitable for publication in PLOS ONE. Congratulations! Your manuscript is now being handed over to our production team.

Kind regards, 

on behalf of

Dr. Ravi Shankar Yerragonda Reddy 

Academic Editor

PLOS ONE